# Deep feedforward functionality by equilibrium-point control in a shallow recurrent network

## Abstract

Recurrent neural network based machine learning systems are typically employed for their sequential functionality in handling time-varying signals, such as for speech processing. However, neurobiologists find recurrent connections in the vision system and debate about equilibrium-point control in the motor system. Thus, we need a deeper understanding of how recurrent dynamics can be exploited to attain combinational stable-input stable-output functionality. Here, we study how a simplified Cohen-Grossberg neural network model can realize combinational multi-input Boolean functionality. We place our problem within the discipline of algebraic geometry, and solve a special case of it using piecewise-linear algebra. We demonstrate a connectance-efficient realization of the parity function as a proof-of-concept. Small-scale systems of this kind can be easily built, say for hobby robotics, as a network of two-terminal devices of resistors and tunnel diodes. Large-scale systems may be energy-efficiently built as an interconnected network of multi-electrode nanoclusters with non-monotonic transport mechanisms.

## 1   Introduction

Shallow recurrent neural networks are being investigated for more context-aware object recognition [25] and brain-like behaviour [23]. They can be more compact (by trading space for time) and are a naturally robust alternative to deep neural networks (which are easily fooled by input perturbations or transformations [18, 29, 1]) when the role of recurrent dynamics is not to produce time-varying output but instead to produce transient (hidden) state-dynamics that facilitate deep, robust and transformation-invariant fixed-input fixed-output functionality. To better engineer such dynamics, we shall study equilibrium-point control, which can be defined as the process of steering to a target in state-space by fixing the input signal, instead of driving it by a continuously varying input signal. Historically, equilibrium-point control [14, 5] was first formulated to provide a plausible solution to the degrees of freedom problem in motor control [3], that is, we mentally represent intermediate destination points rather than a continuum of velocity information required to execute a movement.

Here, we shall focus on using equilibrium-point control to realize multi-input Boolean functionality, in particular the parity function, which is a canonical proxy for nonlinear classification. Theoretical results in circuit complexity are known already for realizing Boolean functionality out of feedforward neural networks, with weighted-sum thresholded binary-output neurons [35]. It has been shown that arbitrary $N$-input Boolean functions can be realized in depth-3 feedforward networks with fewer neurons ($m = \mathcal{O}(2^{N/2})$ instead of the $\mathcal{O}(2^N)$ in total required for depth-2). However, with the advent of nanoelectronics, the size of an artificial neuron has been downscaled to such an extent that it is rather the interconnect wiring that now occupies a greater area in chip design. Thus for a fully-connected deep network, the area scales as the number of interconnects $m^2 = \mathcal{O}(2^N)$. Such a

Submitted to 36th Conference on Neural Information Processing Systems (NeurIPS 2022). Do not distribute.

$\mathcal{O}(2^N)$ scaling law was earlier obtained by Shannon [34] for realizing arbitrary $N$-input Boolean functions by an interconnection of input-controlled switches (or equivalently a feedforward network of 2-input Boolean gates). Thus, unless we employ higher-order neurons [16], we can say that a *Shannon bottleneck* limits the maximum $N$-input Boolean logic realizable in a given area by (nanoscale) feedforward networks. We aim to circumvent this Shannon bottleneck by employing recurrent physical networks. It is known that certain combinational logic functions can be realized by fewer logic gates in a cyclic network than in an acyclic network [31], and with analog signal processing the improvement factor could be even higher.

In the following section, we introduce a state-space model formalism to study equilibrium-point control, and commit to a physically realizable model, and discuss how a general solution for its equilibrium points is a difficult problem in algebraic geometry. Thus, we proceed to idealize the non-monotonic output of the physical system as a piecewise-linear function and solve for the equilibrium points. Finally, a piecewise-quadratic Lyapunov function is obtained for stability analysis and conditions for a unique equilibrium-point are provided.

After the theory, in the results section, we provide a connectance-efficient realization of the parity function. The discussion section puts our results into a broader context and offers avenues for further research. Our objective here is to work at the intersection of nonlinear dynamical systems, neural networks, unconventional neuromorphic hardware, cyclic Boolean circuits, piecewise-linear control systems, and algebraic geometry.

# 2   Theory

## 2.1   State-space model

For equilibrium-point control, in general we have an input vector $\boldsymbol{x}$, a state $s_i(t)$ for $i = 1 : N$, and an output $y$ obtained from a system of equations

$$\dot{s}_i(t) = F_i(\boldsymbol{s}(t), \boldsymbol{x}),\ y = \lim_{t \to \infty} G(\boldsymbol{s}(t)). \tag{1}$$

In this paper, we commit to a physically realizable recurrent network with voltage nodes $s_i$ from $i = 1 : N$, with a capacitive time-constant $\tau_i$, using resistors (of a constant conductance $f_{ij}$) and tunnel diodes (of a voltage-dependent conductance $G_i(s_i)$) as shown in Fig. 1, yielding a state-space model of the form

$$\tau_i \dot{s}_i = x_i - \sum_{j \neq i} f_{ij}(s_i - s_j) - G_i(s_i),\ y = G_1(\hat{s}_1) \tag{2}$$

where $f_{ij} \geq 0$, $G_i$ is a nonlinear passive function such that $G_i(s)s \geq 0$ and $\hat{s}_1 \equiv \lim_{t \to \infty} s_1(t)$ is the stable equilibrium-point if one exists (note: $y(\boldsymbol{x})$ can be multi-valued and depend on the basin of attraction that the initial state $\boldsymbol{s}(0)$ lies in). Brain-scale systems of this kind may be realized by an interconnected network of nanoclusters with non-monotonic transport mechanisms as proposed in [24, Chapter 5]. However, finding suitable network parameters that result in practical functionality remains a challenge. Note that, although not the focus of this work, Eq. (2) can also represent state-space models with noisy rectified-linear units, for which semi-analytical results are known from a computational neuroscience [12] and a machine learning [33] perspective.

## 2.2   Algebraic geometry of the equilibrium points

A study of the set of equilibrium points of a state-space model, $\mathcal{S}_0(\boldsymbol{x}) \equiv \{\boldsymbol{s} \ni F_{1:N}(\boldsymbol{s}, \boldsymbol{x}) = \boldsymbol{0}\}$, can not only help in characterising the stable equilibrium-points $\hat{\boldsymbol{s}} \in \mathcal{S}_* \subseteq \mathcal{S}_0$, but also provide necessary (but not sufficient) conditions in the parameters defining the functions $F_{1:N}$ and $G$, to realize desired equilibrium-point functionality $y(\boldsymbol{x})$. For example, to realize a Boolean function $y : \{0, 1\}^N \to \{0, 1\}$, the following property has to be satisfied:

$$\min_{\boldsymbol{s} \in \mathcal{S}_0(\boldsymbol{x})} G(\boldsymbol{s}) \leq 1 \wedge \max_{\boldsymbol{s} \in \mathcal{S}_0(\boldsymbol{x})} G(\boldsymbol{s}) \geq 0\ \forall \boldsymbol{x} \in \{0, 1\}^N. \tag{3}$$

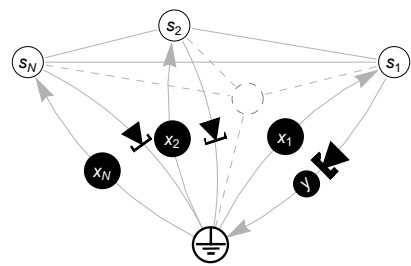

Figure 1: Recurrent physical network corresponding to the state-space model (2) where the inputs $x_{1:N}$ are currents, the states $s_{1:N}$ are voltages, the output $y$ is a measured current, the linear interactions are due to resistors with a conductance $f_{ij}$ between node $i$ and $j$, and nonlinear interactions are due to tunnel diodes from node $i$ to GND with conductance $G_i(s_i)$.

The set of equilibrium points of our recurrent physical network model (2) are the roots of the system of nonlinear equations

$$-f_{i,1:N} \cdot s_{1:N} + G_i(s_i) = x_i \tag{4}$$

where the linear-interaction matrix $f_{N \times N}$ has terms $f_{ii} \equiv -\sum_{j \neq i} f_{ij}$.

Solving the multivariate nonlinear equation (4) is a difficult problem in algebraic geometry, a discipline of mathematics which classically grew around efforts to understand the roots of multivariate polynomials and later metamorphosed by the study of integer-coefficient piecewise-linear functions, with an abstract language that has even recently been applied to explain circuit complexity results of deep feedforward networks [35, 30] through the lens of rational piecewise-linear functions [40].

Algebraic geometry originally dealt with a qualitative approach by geometrical arguments [15], in contrast to a quantitative approach by numerical methods. An example of that kind is Harnack's curve theorem [19] which states that for a 2-D polynomial curve of degree $n$, the maximum number of connected components is $(n^2 - 3n + 4)/2$. Now, with the advent of computer algebra, the roots of multivariate nonlinear equations are studied by the elimination of variables, using techniques such as resultants [13, 38] and Groebner bases [7, 8] for polynomial systems, and as an instance of the linear-complementarity problem [11] or equivalently as absolute-value equations [26] for piecewise-linear systems [37]. However, computer algebra is not scalable for higher dimensions. Thus there is a need to convey the richness in algebraic geometry using analytical expressions. While it is unlikely that analytical expressions may be obtained for any general form of nonlinearity, we may hope that the set of exactly solvable models can be extended well beyond linear equations, a hope banking on our successful experience from other areas of mathematics such as integral calculus [9, section IX] and iterated mappings [39, page 1098].

### 2.2.1 Piecewise-linear algebra

In this paper, we commit to a piecewise-linear analysis by considering

$$G_i(s) = \begin{cases} g_{i1}s & 0 \leq s \leq g_{i2} \\ (g_{i1} + g_{i3})g_{i2} - g_{i3}s & g_{i2} \leq s \leq g_{i2}(1 + \frac{g_{i1}}{g_{i3}}) \end{cases}, \tag{5}$$

where $g_{i1,2,3} > 0$ so that $G_i$ is a triangular peak function in a limited range of $s$, thus defining an idealized negative-differential behaviour. Shifting the state-space about its inflection points as $\boldsymbol{z} \equiv \boldsymbol{s} - \boldsymbol{g}_2$ and then combining (5) with (4) yields

$$x_i = \begin{cases} -f_{i,1:N} \cdot \boldsymbol{z} + g_{i1}z_i - f_{i,1:N} \cdot \boldsymbol{g}_2 + g_{i1}g_{i2} & -g_{i2} \leq z_i \leq 0 \\ -f_{i,1:N} \cdot \boldsymbol{z} - g_{i3}z_i - f_{i,1:N} \cdot \boldsymbol{g}_2 + g_{i1}g_{i2} & 0 \leq z_i \leq g_{i2}(\frac{g_{i1}}{g_{i3}}) \end{cases}, \tag{6}$$

which can be simplified to

$$x_i = -f_{i,1:N} \cdot \boldsymbol{z} + g_{i\ominus}z_i - g_{i\oplus}|z_i| - f_{i,1:N} \cdot \boldsymbol{g}_2 + g_{i1}g_{i2} \tag{7}$$

104    for $-g_{i2} \leq z_i \leq g_{i2}(\frac{g_{i1}}{g_{i3}})$ with $g_{i\ominus} \equiv (g_{i1} - g_{i3})/2$ and $g_{i\oplus} \equiv (g_{i1} + g_{i3})/2$.

105    The system in (7) can be expressed in the absolute-value equation normal form

$$\mathbf{A}z - |z| = b \tag{8}$$

106    with $A_{ij} = (-f_{ij} + \mathrm{I}_{ij}\, g_{i\ominus})/g_{i\oplus}$, $b_i = (x_i + f_{i,1:N} \cdot \boldsymbol{g}_2 + g_{i1}g_{i2})/g_{i\oplus}$ and the bounds

$$-g_2 \leq z \leq g_2 g_1/g_3. \tag{9}$$

107    Similarly, (2) can be expressed as

$$\boldsymbol{\tau}\dot{z} = \boldsymbol{g}_{\oplus}(\boldsymbol{b} - \mathbf{A}z + |z|). \tag{10}$$

Two sufficient conditions are known for the absolute value equation (8) to have a unique solution based on the largest singular value $\sigma_{\min}$ [26] and the spectral radius $\rho$ [32]:

$$\sigma_{\min}(\mathbf{A}) > 1, \tag{11}$$

$$\rho(|\mathbf{A}^{-1}|) < 1. \tag{12}$$

108    However, those are not yet sufficient conditions for a unique equilibrium-point solution for (2) and
109    (10) because the bounds in (9) were not enforced. Thus, we shall proceed to obtain a Lyapunov
110    function to guarantee that a stable equilibrium-point is reached.

## 111   2.3   Lyapunov stability analysis

112    Equilibrium-point stability for large complex systems is not guaranteed in general [17, 27], and the
113    effective dimensionality of stable-input stable-output responses is richly dependent on the parameter
114    space [2]. However, the interaction matrix for our physical system (2) is symmetric, and hence the
115    system is a special case of the Cohen-Grossberg model [10]

$$\dot{s}_i = a_i(s_i)[b_i(s_i) - \sum_{j=1}^{N} c_{ij} d_j(s_j)], \tag{13}$$

with $a_i(s_i) = 1/\tau_i$, $b_i(s_i) = x_i - G_i(s_i)$, $c_{ij} = -f_{ij}$ and $d_j(s_j) = s_j$. Thus, it is known to be globally absolute stable, with a Lyapunov function

$$V = -\sum_i \int_0^{s_i} b_i(u) d_i'(u) \, \mathrm{d}u + \sum_{i,j} \frac{c_{ij}}{2} d_i(s_i) d_j(s_j) \tag{14}$$

$$= \sum_i \left( P_i(s_i) - x_i s_i - \sum_{j>i} f_{ij} s_i s_j - \frac{f_{ii}}{2} s_i^2 \right), \tag{15}$$

$$\text{where output power } P_i(s_i) \equiv \int_0^{s_i} G_i(u) \, \mathrm{d}u. \tag{16}$$

116    Alternatively, since our system (5) is piecewise-linear, a piecewise-quadratic Lyapunov function may
117    be obtained by a piecewise-affine system [22] analysis. While this approach is more powerful and
118    holds even for asymmetric interaction matrices, it also seems to be analytically complex. From another
119    angle, global asymptotic stability [21, Theorem 3] is guaranteed if the Jacobian matrix $\mathbf{J}$ satisfies
120    $J_{ii} + 1/2 \sum_{j \neq i} |J_{ij} + J_{ji}| < 0 \iff G_i'(s_i) > 0$ because in our system $J_{ii} = -\sum_{j \neq i} f_{ij} - G_i'(s_i)$
121    and $J_{ij} = f_{ij}$. Since our network employs non-monotonic functionality, $G_i'(s_i) > 0$ cannot be
122    guaranteed for all reachable states $s_i$, and thus the above criteria is unfortunately inapplicable. Hence,
123    we shall proceed with the Cohen-Grossberg approach.

124    The power function (21) simplifies to

$$P(s) = \begin{cases} \int_0^s g_1 u \, \mathrm{d}u = g_1 s^2/2 & 0 \leq s \leq g_2 \\ g_1 g_2^2/2 + \int_{g_2}^s (g_1 + g_3) g_2 - g_3 u \, \mathrm{d}u & g_2 \leq s \\ = g_1 s^2/2 - g_{\oplus}(s - g_2)^2, \end{cases} \tag{17}$$

125    and using the rectifier function $[x) \equiv \max(x, 0)$ may be expressed conveniently as

$$P(s) = g_1 s^2/2 - g_{\oplus}[s - g_2)^2, \tag{18}$$

126    when the system is within its operational bounds.

# 3  Results

Given the Lyapunov stability result of our system, it is computationally efficient to simulate our state-space model and probe for combinational functionality. Here, we will simulate for the simplest proof-of-concept for deep functionality in a shallow recurrent - solving a parity problem.

Using a cascade of 2-input XOR gates, the $N$-bit parity function can be realized with $N/2 + N/4 + ... + 1 = N - 1$ gates and $2N - 1$ connections. Thus its total cost in area is at least $3N - 2$ units. A minimally-connected network has $N$ input wires, 1 output wire, and $N - 1$ interconnect wires with a total area cost of $2N$ units, assuming that the area occupied by the remaining components is negligible. Thus for $N = 3$, while a conventional digital circuit costs 7 units, our recurrent physical network takes just 6 wiring units.

Our simple model has $N = 3$, $f_{12} = f_{13} = f$, $f_{23} = 0$, $g_{11} = g_1$, $g_{13} = g_3$, $g_{21} = g_{31} = \gamma_1$, $g_{23} = g_{33} = \gamma_3$, $g_{12} = g_2$ and $\gamma_{22} = \gamma_{32} = \gamma_2$. We find from a symbolic evaluation that $\sigma_{\min}(\mathbf{A}) \neq 1/\rho(|\mathbf{A}^{-1}|)$ in general, and conditions for unique stability were not obtainable (which is not surprising due to the $s_2 - s_3$ symmetry). Thus, parity functionality was found by trial-and-error yielding the parameters $\{f = 1.751, g_1 = 1.876, g_2 = g_3 = 0.126, \gamma_1 = 0.876, \gamma_2 = 1.6, \gamma_3 = 0.751\}$ and simulated using Wolfram Mathematica 13 (code in Appendix). When $x_1 = x_2 = x_3 = 1$, the states were forced to transition beyond the bounds in (5), so its range was extended by taking an absolute value. The results are plotted in Fig. 2.

# 4  Discussion

Our result should be seen as a theoretical proof-of-concept and as a motivation for continued research in this area. Future work must extend our simulations to much higher dimensions to serve as a practical demonstration of deep functionality by shallow recurrent networks. Moreover, the theoretical formalism introduced here is not yet fully exploited. We hope to find an analytical method to design functionality out of piecewise-linear Cohen-Grossberg networks.

Our style of reasoning to circumvent the Shannon bottleneck may also be applied to other systems such as networks of coupled oscillators [28]. Our non-modular mode of signal processing, offers an alternative to not just circuit designers, but also to systems biologists who typically understand chemical reaction networks [6] as a composition of modules [20]. While, we have discussed equilibium-point functionality in a state-space model driven by an additive input, it is also worth investigating autonomous systems where the input is set as an initial state. An example is realizing unboundedly-finite parity functions using just a radius-4 cellular automaton [4]. Finally, we hope that this paper can serve as a call to action for neuromorphic engineers to look at physical reservoir computing [36] from another angle, besides temporal input-output functionality.

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

# A  Appendix

Wolfram Mathematica code to reproduce Figure 2.

```
In[◦]:= simulate[f_, g_, γ_] := (sys = NonlinearStateSpaceModel[{
        {x1 - (2 f) s1 + f (s2 + s3) - Abs[g[[1]] * s1 - (g[[1]] + g[[3]]) Ramp[s1 - g[[2]]]],
         x2 - (f) s2 + f (s1) - Abs[γ[[1]] s2 - (γ[[1]] + γ[[3]]) Ramp[s2 - γ[[2]]]],
         x3 - (f) s3 + f (s1) - Abs[γ[[1]] s3 - (γ[[1]] + γ[[3]]) Ramp[s3 - γ[[2]]]]},
        {x1, x2, x3, Xor[x1, x2, x3], s1, s2, s3, y = Abs[g[[1]] s1 - (g[[1]] + g[[3]]) Ramp[s1 - g[[2]]]], HeavisideTheta[y - .15]}
      }, {s1, s2, s3}, {x1, x2, x3}];
    inputs = {.5 - .5 * SquareWave[t / 50], .5 - .5 * SquareWave[t / 100], .5 - .5 * SquareWave[t / 200]};
    out = OutputResponse[{sys, {0, 0}}, inputs, {t, 0, 200}];
   GraphicsColumn@Table[Plot[out[[i]], {t, 0, 200}, PlotRange → All, Ticks → {Automatic, {0, 1 / 5, 1, 2}}], {i, 9}])

   simulate[1.751, {1.876, .126, .126}, {.876, 1.6, .751}]
```

