# OpenReview forum: "Deep feedforward functionality by equilibrium-point control in a shallow recurrent network."
_NeurIPS.cc/2022/Conference — NeurIPS 2022 Submitted_

### Official Review · Reviewer_NkSi · 2022-07-03

**Rating:** 2
**Confidence:** 1
**Soundness:** 2 fair
**Presentation:** 1 poor
**Contribution:** 1 poor

**Summary:**

A feedforward logic circuit follows certain scaling constraints (for example, the number of layers or logic switches required to compute some function). However, a cyclic (analogue) network can, in some cases, avoid these limits.

This work introduces an approach to using analogue, cyclic networks to compute the parity (highly non-linear) logic function. They analyze and simulate (to find the parameters) this for a N=3 problem.

**Questions:**

- What are the main contributions of this work?

- Could you explain more clearly what you expect NeurIPS audiences to take away from this paper and which parts of the community are most likely to be interested.

**Strengths And Weaknesses:**

Strengths:
- Using recurrent functionality to reduce circuit sizes in an interesting problem (although the practical realizations make be non-trivial at scale).

Weaknesses:
- Paper is very hard to follow and makes a lot of assumptions on prior knowledge (and there is plenty of remaining space to include more background, its only 5 pages right now).

- Its not clear what the main contributions of this work are.

- The scale of this work is trivial, and the solution was found numerically. This approach is much more interesting for much larger scale.

---

> ### Author Response · Authors · 2022-07-28
> **Request the reviewer to ask non-generic questions**
>
> The answers to this reviewer's questions are found on reading the abstract, introduction, discussion parts of the paper and mentioned in Checklist 1. (a).  The purpose of a conference is to facilitate discussion among expert or open-minded researchers, inspire fresh thinking, and not write an expository survey paper. Hope the summary of this work by Reviewer jxFt is able to satisfy you.
>
> That the reviewer finds the work interesting for much larger scales is welcome. This work is a step in that direction!

---

### Official Review · Reviewer_jxFt · 2022-07-05

**Rating:** 7
**Confidence:** 2
**Soundness:** 4 excellent
**Presentation:** 2 fair
**Contribution:** 3 good

**Summary:**

The idea of this paper is to limit the number of connections in a physical neural network that are required to realize $N$-input Boolean functions, while focusing on the parity functions as a proof-of-concept. In feed-forward networks the Shannon bottleneck demands that there are $O(2^N)$ interconnections between neurons to realize the desired Boolean functions. The authors try to circumvent this bottleneck by using recurrent physical networks that have been shown to be able to realize certain functions with fewer units.

After defining the state-space model of the recurrent physical network, the authors are looking for sufficient conditions of the conductance function that allow for the existence of its equilibrium points in order to prove that a stable output activation can be reached. This is achieved by committing to analyzing a conductance function that is piecewise-linear algebraically. Further, due to the nature of the proposed state-space model, the system becomes a special case of the Cohen-Grossberg model which can be shown to be globally absolute stable using a Lyapunov stability analysis.

The authors use the result of the stability analysis to simulate a shallow recurrent neural network that solves the parity problem.

**Questions:**

Please provide more information on how you think the analysis can be scaled up to more realistic problems.

**Limitations:**

The authors consider the work as a proof-of-concept and describe several opportunities for future work. An analysis on potential negative societal impact has been omitted, which is reasonable given the theoretical nature of the paper.

**Strengths And Weaknesses:**

The paper shows an interesting and scientifically important direction for future physical neural network designs, which potentially could allow for many more efficient physical neural network implementations in practice. The authors provide a sound theoretical analysis of their proposed system.

The writing is very compact but of high quality. Readability could be improved by providing more background information on physical neural networks, as these are not particular common in the general machine learning community.
In general I would appreciate a more detailed descriptions of their methods and results - the paper is very short in its current form and I see a lot of opportunity to improve clarity without getting close to the page limit.
I would also like to see the authors putting the paper better into context with other publications by citing related work in the field.

---

> ### Author Response · Authors · 2022-08-02
> **Scaling up needs more theory**
>
> Thank you for your thoughtful summary and clear evaluation about the strengths and weaknesses of this work. It would also be great if you have suggestions for related work that is missing (the paper has 40 references spanning multiple areas, but no reference to other work using recurrent neural networks for combinational input-output functionality was found).
>
> Scaling up the analysis to more realistic problems may either be done empirically or by building up on the theory introduced in this work. For example, we could devise some analytical form for the interaction-matrix $\textsf{\textbf{A}}$ in the system of absolute-value equations (8), such that the system is uniquely solvable. Next, we need to check what families of such interaction matrices are physically realizable and find expressions for them. This is not an easy task and clever constructions are needed.

---

### Official Review · Reviewer_HA6S · 2022-07-08

**Rating:** 6
**Confidence:** 1
**Soundness:** 3 good
**Presentation:** 2 fair
**Contribution:** 3 good

**Summary:**

This paper analyses the dynamics of recurrent neural networks and studies the equilibrium points to produce stable combinatorial output.
The authors propose a physical model amenable for analysis with algebraic geometry, solve for the equilibrium points under the assumption of piecewise-linear non-monotonic output, and perform Lyapunov stability analysis. The experimental phase validates the theoretical result by realizing the parity function.

**Questions:**

- The current work assumes piecewise-linear output that fits well combinatorial boolean functions. However, a broader class of functions can be represented or approximated as piecewise-linear (e.g., absolute value function). Is the current work limited to boolean function in its assumptions?
- I would recommend the authors extend the manuscript with more motivations, background notions, running examples, and material to make the current work accessible to a broader audience. In particular, clearly communicating the goal and its applicability is essential. Otherwise, the submission of theoretical work, even if providing novel and strong results, could not be appreciated by the readers.
- Section 2.2 about algebraic geometry is partially unclear. For example, equations (3) and (5) define the properties and assumptions of the problem. It would be nice to extend them with some visualizations of a running example.

**Limitations:**

The main limitations of this work are the clarity in communicating the theoretical results and the limited experimental phase.
As an extension, it might be worth validating the architecture scalability results by conducting experiments for varying N.

**Strengths And Weaknesses:**

- *Significance*: This work advances the theoretical analysis of recurrent neural network dynamics, which might significantly contribute to the community and help design more efficient architectures. However, the paper is very concise, and part of the details is difficult to grasp at first read. In the Introduction, the final aim is not sufficiently stressed, and the discussion about the need for more efficient architectures is limited to a paragraph. The experimental phase consists of a simple experiment on standard parity function with few input dimensions. Nevertheless, it is a nice proof of concept to validate the proposed theoretical result.

- *Originality*: The paper adequately mentions prior works along the text and explains how the contribution differs from them. However, a separate section on related work might be helpful to compare the approach with prior works easily.
The analysis of recurrent neural networks is novel, and this work uses appropriate analysis techniques from algebraic geometry and Lyapunov stability.

- *Clarity*: The paper is well written but difficult to follow. It assumes a strong knowledge from the reader, presenting only the principal results and citing prior works from the literature. Considering it is a theoretical work, I would have appreciated a section to provide a minimal technical background for the readers. To this aim, the authors could have used all the allowed pages and provided additional content in the appendix.

- *Quality*: The submission looks technically solid, and the analysis techniques look appropriate. However, the soundness of the technical claims is beyond my knowledge.


### Minor remarks

- Figure 2 has no labels on the axis. Moreover, the vertical axis shows different lengths (see first and second plots) and scales (see last and second-last plots) for the various quantities visualized.

---

> ### Author Response · Authors · 2022-08-02
> **Given resource-constraints, complex concepts could be made clearer in a slide-show.**
>
> 1. Yes, a broad class of functions can be approximated as piecewise-linear. The focus here is on the parity function because it is highly-nonlinear (of maximal threshold order [41]). Thank you for acknowledging that this is already a nice proof-of-concept.
>
> 2.  Empirical results based on varying N can be provided. However, it is more insightful to first build on the theory as suggested in the response to Reviewer jxFt. If analytical expressions are available, then it would be possible to optimize the abundance of nonlinear combinational capacity in recurrent neuronal networks (so far, only theoretical estimates in terms of the neuronal capacity [42] of fully connected recurrent neural networks is known).
>
> 3. Thank you for the minor remark, it is indeed an unconventional choice to put the y-axis labels of Figure 2 on the top of each plot and the shared x-axis mentioned in the figure caption to be from 0 to 200 time units. The focus here is on a qualitative rather than quantitative interpretation.
>
> **Extra references**
>
> [41] Wang C, Williams AC. The threshold order of a Boolean function. Discrete Applied Mathematics. 1991 Mar 11;31(1):51-69.
>
> [42] Baldi P, Vershynin R. On neuronal capacity. Advances in Neural Information Processing Systems. 2018;31.

---

### Official Review · Reviewer_ZLGu · 2022-07-11

**Rating:** 4
**Confidence:** 2
**Soundness:** 3 good
**Presentation:** 3 good
**Contribution:** 2 fair

**Summary:**

To reduce tne number of connections in a neural network needed to compute a Boolean function, the authors use a recurrent neural network instead. Given input x, the RNN must settle to an equilibrium, and the first node of the network is transformed to give the output. The authors use the Cohen-Grossberg theorem to show stability if the connections are symmetric. They obtain requisite connections for the N-bit parity function by trial and error.


**Questions:**

Would say using backprop through time to learn the symmetric connections be possible while maintaining the equilibrium guarantees (during learning and at the end)? Or could you provide further insight into learning or constructing such networks. If this or some other constructive algorithm was proposed showing the connections to your theoretical methods, I would be happy to improve my rating.

**Limitations:**

The work is still in a rather nascent stage and these issues do not arise yet.


**Strengths And Weaknesses:**

The authors motivate the work well suggesting that connectivity is the bottleneck for neuromorphic hardware rather than neuron numbers, and thus suggest recurrent neural networks that reach equilibrium as more efficient to compute static (binary) input-output mappings. They show some tantalizing connections to algebraic geometry, and try to do a piecewise linear analysis. To my inderstanding, these yield only necessary but not sufficient conditions. They instead show Lyapunov stability of the system assuming symmetric connections. The authors point out some very interesting connections that have the potential to lead to interesting results. However, the authors have been unable to develop these further theoretically. The N-bit parity function was found by trial and error and the authors do not provide a constructive algorithm to learn more general binary input-output transforms. Thus the experimental work is also rather limited and does not prescribe how these networks may be constructed or learned.

---

> ### Author Response · Authors · 2022-08-02
> **Conventional methods will likely not work**
>
> Backprop through time to learn the weights of the interaction matrix is an interesting idea, but will likely not scale up.
> Please check the response to Reviewer jxFt for one possible constructive (theory-based) method.
>
> As mentioned in the conclusion of this paper (perhaps this should also have been in the abstract?), reservoir computing could be a way to exploit deep functionality out of shallow physical recurrent networks. In this paradigm, only the output layer is meant to be learnt (similar to Rosenblatt's perceptrons of the 1960's where only the association to response layer was meant to be learnt- so in some sense going back to the origins of neural networks research, but unconventionally). To demonstrate that physical networks would result in scalable reservoirs, analysis for large N is to be done (as noted in the response pt. 2 to Reviewer HA6S).

---

### Public Comment · Authors · 2022-11-09
**Author response to metareview**

I find the metareview strong in diplomacy but weak in science. Thanks for acknowledging that my submission is highly interdisciplinary, potentially of high impact, technically sound and further providing a reference. Coincidentally, I have read the work of Gu. et al before (even made a Wolfram demo inspired by their thought-provoking work https://demonstrations.wolfram.com/OrthonormalPolynomialsUnderDifferentInnerProductMeasures/). However, they deal with sequential functionality out of linear-discrete-recurrent systems, while I deal with combinational functionality out of nonlinear-continuous-recurrent systems. I can comment a lot more about this, but it is a complete distraction to the Cohen-Grossberg notion of stability that I deal with here.

Finally, I thank those reviewers who read the manuscript carefully and provided constructive feedback.

---

### Meta-Review · Area_Chair_xC5s · 2022-08-25

**Recommendation:** Reject
**Confidence:** Certain

**Metareview:**

We had quite a bit of discussion on this paper. I read the paper and agree with some of the discussions that the paper in its current form might not attract interest in the community due to the following reasons:

- While it is highly interdisciplinary and potentially of high impact, the authors did not manage to connect all topics to tell a concise story on the use of equilibrium-point control theory in making a new recurrent neural network model. Here are some final comments from our discussions with the reviewers:
    - While the content looks technically sound, we haven't seen from the authors the expected revision to improve the paper. We were expecting a better introduction to the problem and further discussion to convey the contribution.
    - We think that it would be in the interest of the authors to extend the paper with clarifications and more background information such that a larger audience will be able to learn something from their work.
    - We believe that this issue is fixable as there is ample space to add additional background/introduction to explain the problem/relevance. However, the authors did not manage to convince the reviewers how they are going to address this during the discussion period.

- The authors could elaborate much more in detail on all topics involved to build a better understanding first and convey their contribution and impact better.

- Also, there are recent advances in state-space models and their use as expressive representation learning algorithms within the community which are disregarded in the paper. In particular, the stability condition, memorization, the efficiency of computing transition matrices (which are very relevant to this paper), and more properties are discussed the last few years. Here is an example:

[1] Gu et al. Efficiently modeling long sequences with structured state spaces, ICLR 2022 https://arxiv.org/abs/2111.00396

Based on these points, I vote for the rejection of this paper.

**Award:**

No

---

### Decision · Program_Chairs · 2022-09-14

Reject